# Nine Years of African Swine Fever in Poland

**DOI:** 10.3390/v15122325

**Published:** 2023-11-27

**Authors:** Mateusz Kruszyński, Kacper Śróda, Małgorzata Juszkiewicz, Dominika Siuda, Monika Olszewska, Grzegorz Woźniakowski

**Affiliations:** 1County Veterinary Inspectorate, Stanisława Dubois 3, 46-100 Namyslow, Poland; jakosc.kruszynski@gmail.com; 2Institute of Veterinary Medicine, Nicolaus Copernicus University in Toruń, Lwowska 1, 87-100 Torun, Poland; 304286@stud.umk.pl; 3Department of Swine Diseases, National veterinary Research Institute, Partyzanotw 57 Avenue, 24-100 Pulawy, Poland; malgorzata.juszkiewicz@piwet.pulawy.pl; 4Academia Copernicana Interdisciplinary Doctoral School, Bojarskiego 1, 87-100 Torun, Poland; d.siuda@doktorant.umk.pl; 5Department of Infectious, Invasive Diseases and Veterinary Administration, Institute of Veterinary Medicine, Nicolaus Copernicus University in Toruń, Lwowska 1, 87-100 Torun, Poland; olszewska-tomczyk@umk.pl

**Keywords:** African swine fever, ASF, spread, eradication measures, Poland, nine years

## Abstract

(1) Background: African swine fever (ASF) is a highly contagious and fatal haemorrhagic disease in domestic pigs and wild boars, causing significant economic loss to the swine industry in the European Union. The genotype II of African swine fever has spread in many European countries since the virus was detected in 2007 in Georgia. In Poland, the genotype II of the ASF virus was confirmed on 17 February 2014 in the eastern part of the country and appeared to have been transmitted to Poland from Belarus. Poland has been particularly affected by ASF epidemics in the last decade, resulting in a significant decline in the Polish pig population. Wild boars are the main reservoir of the African swine fever virus (ASFV), but human activities such as transportation and illegal animal trade are the primary reasons for the long-distance transmission of the disease. (2) Conclusions: During the nine years of ASF in Poland, multiple measures have been taken to prevent the spread of the virus among the wild boar population via the passive and active surveillance of these animals. With regard to pig farms, the only effective measure for preventing the spread of ASF is the efficient enforcement by state authorities of the biosecurity standards and the farmers’ compliance with them.

## 1. Introduction

African swine fever (ASF) is a viral, contagious and often fatal disease in domestic pigs (*Sus domestica)* and wild boars (*Sus scrofa*) [1]. ASF was first described in 1921 in Kenya as an acute haemorrhagic fever with an extremely high mortality rate (up to 100%) in domestic pigs (*Sus domestica*). Previous studies have shown that the source of infection was the population of warthogs (*Phacochoerus africanus*), which were not susceptible to the virus themselves; however, they acted as a reservoir, causing infection in domestic pigs [2].

ASFV is a large, enveloped virus with icosahedral symmetry of the capsid, with a genome that constitutes a double-stranded DNA of 170–190 kbp in length. The virus replication occurs in the cytoplasm of the infected animal cell. However, the cell nucleus is more equally the site of viral DNA synthesis at the early stage [3,4]. Initially, ASFV was classified as a member of the family *Iridoviridae* [5], but in 1999, it was placed into the *Asfaviridae* family [6]. So far, 24 different genotypes of ASFV have been described [7]. Currently, there are two main clusters of ASFV infections in Europe: Sardinia, where the disease was first found in 1978 and caused by the ASFV genotype I strain, and northeastern Europe, affected by the ASFV genotype II strain. This is a particularly virulent strain that causes the acute form of ASF, with a high mortality rate of up to 100% in both wild boar and domestic pigs [8]. The main sources of ASFV are infected domestic pigs and wild boars, as well as contaminated feed, water, semen, pork, farm personnel, vehicles, or fomites [9]. Maintaining high standards of biosecurity measures, including disinfection, may reduce many potential sources of infection. Ticks of the *Ornithodoros* genus in endemic areas have been shown to be a frequent vector of the virus, participating in its transmission [10,11]. Young warthogs bitten by the ticks become infected with ASFV and, as a result, act as vectors themselves without showing any clinical symptoms. The role of blood-sucking flies of the *Tabanidae* family in virus transmission has also been described. However, the extent of their influence or role is not fully understood or proven [11,12]. 

ASF first appeared in the European continent in 1957 in Portugal, and it is widely believed that the source responsible for the spread of the pathogen in Europe came directly from western Africa (Angola). Then, ASFV was transferred to different parts of Europe and Asia, reaching Spain and other European countries such as France, Italy, Malta, Belgium and the Netherlands. However, in Spain and Portugal, the virus remained in its endemic form. In the 1970s, ASF was also recorded in the American continent, including Cuba, Brazil, the Dominican Republic and Haiti. In 1998, it reached Madagascar [13,14]. A worldwide epidemic of ASF began in June 2007 in Georgia. The disease is suspected to have spread within the country because of the improper disposal of food waste from transport near the Black Sea port of Poti. Therefore, ASFV quickly spread into neighbouring countries, affecting domestic pigs and wild boars in the Czech Republic and the Russian Federation (2007) as well as Azerbaijan (2008) [15]. In August 2012, an outbreak of ASF was detected in the south-eastern part of Ukraine [16], while in June 2013, the Belarusian veterinary authorities confirmed the detection of the ASF virus in the village of Czapuń, Grodno region, 170 km from the Polish border. Another ASF case was also confirmed in the area of north-eastern Belarus, in Vitebsk, 450 km from the Polish border [17]. In 2014, ASFV reached the European Union—the first outbreak of ASFV in wild boar was confirmed in January in Lithuania, February in Poland, followed by Latvia and Estonia. It should be mentioned that, in all the Baltic countries, the cause of ASF was the highly pathogenic ASFV genotype II [18]. Over the following years, the virus gradually spread to other European countries, including the Czech Republic and Romania (2017), Bulgaria, Hungary, Belgium (2018), Slovakia (2019), Greece and Germany (2020) and Italy and North Macedonia (2022). In August 2018, ASF was detected in the world’s largest pig producer, China, from where it quickly spread to southeast Asia. ASFV has also been confirmed in Oceania (Papua New Guinea), India, Haiti and the Dominican Republic [7,19]. Currently, the disease shows a continuous spread in recently affected areas, including Germany, Italy, Slovakia, Serbia, Croatia and Greece.

## 2. The Beginning of ASF Spread in Poland

The population of domestic pigs in Poland in 2023 is 8.3 mln, distributed in 50,000 pig farms. The average size of the herd is 165 pigs. With regard to wild boar, the highest density of over 2–4 wild boar per square km is in western Poland, and the overall estimated population is approximately 67,000.

The news of the ASFV infections from Eastern European countries prompted the implementation of prevention campaigns and programs. In Poland, monitoring for the ASF virus started as early as 2011. Monitoring activities were carried out in a 40 km wide border area (along the border with the Kaliningrad Oblast) and along the eastern border with Ukraine, Belarus and Lithuania, covering parts of Warmińsko–Mazurskie, Podlaskie, Lubelskie, Mazowieckie and Podkarpackie voivodeships. Samples were collected from fallen and shot wild boar based on risk analysis. From July 2013, samples were taken in the area located in Zone I, and from October 2013, in the area located in Zones II and III. All the monitoring activities were carried out in accordance with a programme aimed at the early detection of infections with the virus-causing ASF. During the two years, 15,187 samples were collected and tested in the designated zones; most of them originated from wild boar. All samples were negative for ASFV. In order to increase knowledge of the risk of ASF in the territory of Poland, in 2013, simulations of the ASF outbreak were carried out. It was intended to test the procedures and liaison between the Chief Veterinary Officer and the Provincial and District Veterinary Inspectorates as well as the Minister of Agriculture and Rural Development in the event of an ASF outbreak [20,21,22].

The first ASF case in Poland was confirmed in the second half of February 2014 by the State Veterinary Research Institute in Puławy (National Reference Laboratory). The DNA sample came from the carcass of a dead wild boar found near the village of Grzybowszczyzna (Szudziałowo Municipality), Podlaskie Voivodeship, about 900 metres from the border with Belarus. The second case of ASF in Poland was found on 17 February 2014 near the village of Kruszyniany, Podlaskie Voivodeship, Sokółka County, near the border with Belarus. The sample was taken from a wild boar carcass, and the distance between these two cases was about 15 km [20,23,24,25].

As of 11 September 2014, 14 outbreaks of ASF in wild boar had been confirmed in 4 municipalities of Podlaskie Voivodeship, in the districts of Sokółka and Białystok County. The distance separating the places where the wild boars were found at the border with Belarus was no more than 9 km. Positive results were found in 32 dead animals of different ages; moreover, additional tests, including sequencing, confirmed that the strain of the virus isolated from these wild boars belonged to genotype II and confirmed its affinity with a strain circulating in the Belarusian, Ukrainian and Lithuanian territories [26].

Five days after the diagnosis of the two ASF cases, blood samples from 623 pigs were tested on 118 farms in 57 different locations. At the same time, within a 40 km radius of the confirmed ASF outbreaks in wild boars, passive monitoring was carried out by searching for the carcasses of these animals; in a similar area, active monitoring was carried out, i.e., shooting of wild boars [22].

## 3. The Spread of the ASF Virus in Eastern Poland

The following numbers of outbreaks were confirmed in wild boars from 2014 to the beginning of October 2023: 30, 53, 80, 741, 2443, 2447, 4156, 3214, 2113 and 2329. In domestic pigs from 2014 to 2023, the following numbers of outbreaks were found: 2, 1, 20, 81, 109, 48, 103, 123, 14 and 30 [27]. This is a recurring phenomenon, as ASF in the pig population occurs seasonally, ordinarily in summer and spring [1,25,28] (Figure 1A,B).

Since 2014, the ASF virus has broken down further territorial barriers and consistently spread throughout the country. The radical depopulation of wild boar and the implementation of biosecurity protocol on pig farms are both crucial measures to reduce the rate of spread of the ASF virus [29].

Initially, the rate of the spread of the ASF virus was relatively slow among the wild boar population, oscillating between 10 and 12 km per year westwards and southwards. The situation changed in 2016 when a total of 17 outbreaks in pigs were identified in Podlaskie, Lubelskie and Mazowieckie Voivodeships. At that time, a hypothesis was put forward that humans were responsible for the transmission of ASF disease further into the Podlaskie Voivodeship. The introduction of the ASF virus into the wild boar population of Mońki County took place by burying the carcasses of dead pigs in the forest. The lack of adequate controls on the transport of pigs and illegal transport of animals was also one of the reasons for the further transmission of the virus over longer distances. A trader selling ASF virus-infected pigs, which originated from the Zambrów district of Brzeźnica, contributed to the outbreak of the disease in the Wysokie Mazowieckie County. A farmer from this town sold piglets without proper documentation [19,28,30,31,32].

The clinical picture in infected domestic pig herds included elevated body temperature as well as redness (cyanosis) of the skin, particularly the ears. During the post mortem examination, an extreme enlargement of the spleen was identified (Figure 2).

The evidence of the major human role in the long-distance transmission of ASFV was provided by the outbreak of the disease in wild boars at the end of 2017 near Warsaw and Piaseczno, which were more than 100 km away from the restricted zones in eastern Poland. On 17 November 2017, the Chief Veterinary Officer announced the presence of the ASF virus in two wild boars found in the municipality of Nieporęt and in the municipality of Jabłonna (Mazowieckie Voivodeship, Legionowo County). The first positive case was a wild boar found dead in a forest while the other had fallen because of a car accident. The crossing of the natural boundary of ASF disease outbreaks, which was the Vistula River, occurred after the confirmation of another positive case by the Chief Veterinary Officer on 22 November this year—a wild boar found in the eastern Warsaw district in the Izabelin commune. Subsequently, further outbreaks of ASF among wild boars were reported in Piaseczno, Nowy Dwór Mazowiecki and in the Polish capital, Warsaw. The total number of confirmed ASF outbreaks in wild boars in the Mazowieckie Voivodeship was 78. In the same year, the ASF virus reached the north of the country—the Warmińso–Mazurskie Voivodeship. The carcass of an ASFV-positive wild boar was found in this province near the village of Skrzypki, in the Ełk district, 58 km from the Polish–Belarusian border.

The year 2018 brought an increase in the number of detected outbreaks of ASFV in the wild boar population. The total number of positive results this year was higher than the sum of all outbreaks recorded in previous years. A very similar situation occurred in Lithuania, Latvia and Estonia [18]. At the beginning of 2018 in Poland, further outbreaks were confirmed in the Warmińsko–Mazurskie Voivodeship, near the border with the Kaliningrad region. During the summer of 2018, a further eight ASF outbreaks were found in pigs in Podkarpackie Voivodeship, in the municipality of Cieszanów [17].

2019 marked an increase in identified ASF outbreaks in Poland and the spread of the virus into the western part of Poland. A total of 2525 outbreaks were reported at the time, including 2477 outbreaks in wild boar and 48 outbreaks in pigs. The highest number of outbreaks in wild boar was reported in the Mazowieckie [923], Warmińsko–Mazurskie (762), Lubelskie (569) and Podlaskie (105) voivodeships. The number of pigs culled was 8016 (8012 pigs on one farm in the municipality of Orla) and Warmińsko–Mazurskie Voivodeship [20], where a total of 20,570 pigs were culled [33] (Figure 3). 

In 2020, the first outbreaks of ASF in wild boar were detected in two provinces (Pomorskie and Zachodniopomorskie voivodeships), where ASF had not previously occurred. The rapid spread of the diseases to other territories of the country was associated with a doubling of the number of confirmed outbreaks in wild boars compared to 2019. The highest number of positive results for ASFV in wild boars was found during passive surveillance involving the search for the remains of dead animals. For comparison, during active surveillance (shooting) conducted in 2020, about 150 times fewer positive cases were confirmed, demonstrating a clear advantage of the effectiveness of passive surveillance over active surveillance [18,30,34,35,36,37,38,39].

## 4. ASFV in Western Poland and Other Provinces in the Country

Initially, ASF was found only in the eastern part of Poland in Podlaskie, Lubelskie, Warmińsko–Mazurskie, Mazowieckie, Świętokrzyskie and Podkarpackie voivodeships. The situation changed in 2019 when the first case of the disease was confirmed in western Poland. In November 2019, in Lubuskie Voivodeship, a dead wild boar was found in Sława, Wschowa County. The positive case from Lubuskie Voivodeship cannot be linked epidemiologically to the previously reported cases of ASF in Poland, as it was located 300 km away from the affected areas. This situation indicates that the virus moved into western Poland by human factor rather than as a result of wild boar migration. In November 2019, ASFV was confirmed in Dolnośląskie Voivodeship (Głogów County, Kotla Municipality), 10 km from the previously confirmed first case in the western part of the country. Over the following months, the virus spread to further areas in western Poland.

As a result of monitoring activities, a total of 130 outbreaks of ASF in dead wild boars had been confirmed by December 2019 out of approximately eight counties located in Wielkopolskie, Dolnośląskie and Lubuskie voivodeships. Based on the data from the forest districts as of March 2019, it was estimated that the number of wild boars oscillated in the range of 0.27–0.52 boars/km^2^ in the west, and the incidence of ASF in wild boars ranged from 80–100%. As an additional security measure, a fence has been erected to limit the migration of wild boars in western Poland. The migration of wild boar near the Polish-German border was closely monitored—a total of 7125 samples were collected; however, all of them were negative [29,31,40,41].

It should be mentioned that the first reported outbreak of ASF in western Poland was about 79 km from the border with Germany, which increased the risk of introducing ASF into a neighbouring country through the migration of infected wild boars. Germany had introduced preventive measures, such as the increased monitoring of wild boars and domestic pigs and the construction of a fence along the Polish border to prevent wild boar migration into Brandenburg. The applied measures to prevent the spread of the virus introduced by Germany were unsuccessful. In September 2020, Germany’s Agriculture Minister confirmed the first case of ASF in a fallen boar found in the Spree-Neiße district of Brandenburg [34,35,42,43].

The ASF virus 2020 was introduced into pig farms located in Lubuskie, Dolnośląskie and Wielkopolskie voivodeships, causing huge economic losses in pig production. At the time, 104 outbreaks of ASF among pigs were reported nationwide, with six outbreaks in pigs in Wielkopolska, four in Lubuskie Voivodeship and two in Dolnośląskie Voivodeship. The first outbreak of ASFV in pigs in 2020 was announced on March 20 in a farm of 27,908 animals in Niedoradz, Lubuskie. On 5 April, another ASF outbreak was reported in Więckowice, Poznań County, Wielkopolska. The farm had 10,074 pigs, and the infection most likely occurred as a result of the importation of infected piglets purchased from a farm in Niedoradz. In June of the same year, the first outbreak of ASFV in Dolnośląskie Voivodeship was found on a farm of 23 pigs. The likely source of the infection was a dead wild boar found near the farm in Dalkowo, while humans were most likely the vector, introducing the virus into the pig population. The second case of ASF confirmed on 27 August 2020, occurred on a farm in the village of Wierzchowice, Dolnośląskie Voivodeship, located about 7 km from the previous outbreak.

The year 2021 was a record year for ASFV outbreaks in pig farms, while fewer outbreaks were reported in wild boars compared to 2020. The largest number of outbreaks in pig farms was confirmed in Podkarpackie (56) Warmińsko–Mazurskie (19) and Dolnośląskie (11) voivodeships. The largest losses in pig production were recorded in Lubuskie Voivodeship, in the village of Niedźwiady, Świebodzin County. On 17 March 2021, an outbreak of ASFV was detected in a large farm with 16,566 animals. In the same year, the virus entered pig farms in new areas of Łódzkie, Świętokrzyskie and Małopolskie Voivodeships. In all likelihood, it can be concluded that the reason for the introduction of the virus into pig farms was human, as the detection of an ASF outbreak on the farm was not preceded by an outbreak in the local wild boar population. The first outbreak of ASF in a pig farm in Łódź Voivodeship [Tuszyn Municipality, eastern part of Łódź County], where 149 animals were kept, was announced on 19 June. Within 2 days, two more outbreaks of ASF were announced in the county of Wieruszów. According to the District Veterinary Officer, this outbreak followed the occurrence of an outbreak located in Wieruszów County. In the same year, from June to September, a total of six outbreaks of ASF in swine farms were announced in Małopolskie Voivodeship. The first outbreak was identified on 21 July in the village of Skrzyszków, and it was further confirmed that the vector of the disease was a human since ASF had not been confirmed among the wild boar population in this region. The virus was most likely transmitted to Małopolskie from Mielec County (Podkarpackie Voivodeship), which was heavily affected by the ASF virus in the year described (196 outbreaks in wild boars and 55 outbreaks in pigs) [44]. The first ASF outbreak in Świętokrzyskie Voivodeship was confirmed on 14 October in Staszów County, with a total of 7 outbreaks in pigs confirmed in this province in 2021. The epizootic investigation confirmed that the virus entered the pig farm as a result of human transmission from Mielec County, located in the Podkarpackie Voivodeship.

The year 2022 was characterised by a significant drop in ASF outbreaks across the European Union. ASF in wild boars in the EU amounted to 7282 cases, which is a 40% decrease compared to 2021. Poland then registered the highest number of ASF outbreaks in wild boars in the entire EU (2152 outbreaks). However, there was a 33% year-on-year decline in this case. The largest number of positive cases in wild boars was recorded in Dolnośląskie, Lubuskie, Warmińsko–Mazurskie and Wielkopolskie Voivodeships. Germany ranked right after Poland with 1628 ASF outbreaks in wild boars, followed by Latvia, Hungary and Slovakia. In 2022, in the Czech Republic, after 4 years without the presence of the disease, ASFV infection was confirmed in a dead piglet found near the border with Poland. A similar situation affected Italy, although the country was free of ASF, except for the endemic form of the disease in Sardinia (genotype I ASFV).

Similarly, an almost 80% decrease in ASF outbreaks in domestic pigs was reported in the European Union. The largest number of outbreaks in pigs was registered in Romania (316), Serbia (103) and North Macedonia (30). Poland confirmed 14 outbreaks of ASF in 5 voivodeships, a decrease of 89% compared to 2021. In 2022, the highest number of cases was reported in Wielkopolskie Voivodeship, where seven outbreaks were announced, with a total number of pigs eradicated ex officio amounting to 2790. The other voivodeships where pig outbreaks were announced included Zachodniopomorskie (2), Lubuskie (1), Dolnośląskie (2) and Warmińsko–Mazurskie (2).

Currently, Poland has 2289 outbreaks of ASFV in wild boars (as of 25 October 2023); compared to other European countries, we have the highest number of ASF outbreaks in wild boars [44,45]. They have been reported in 11 voivodeships: Zachodniopomorskie, Pomorskie, Lubuskie, Wielkopolskie, Dolnośląskie, Opolskie, Podkarpackie, Lubelskie, Mazowieckie, Podlaskie and Warmińsko–Mazurskie voivodeships (Figure 4).

Other countries in Europe where large numbers of ASF are still being reported in wild boars are Germany, Italy and Slovakia.

No outbreaks of ASFV in pigs have been reported in Poland. The current epidemiological situation of ASF in Poland, along with marked restricted areas, is shown in Figure 1. In European countries, ASF outbreaks in pigs have been reported in Romania, Serbia, Moldova, Ukraine and Germany [45,46].

Poland is a country that still struggles with the problem of ASF. It should be mentioned that a significant part of Poland is still free of the virus [19,34], so it is important to take active measures to prevent the spread of the disease, such as monitoring the wild boar population, adhering to the biosecurity measures with the use of disinfectants with confirmed efficacy on pig farms, but also increasing the awareness of people, veterinarians and employees of the farms in question to combat the ASFV. 

The Supreme Audit Office (NIK), analysing the situation of combating and preventing the spread of ASF in the years 2019–2021, issued a statement identifying inconsistencies in supervision by the Veterinary Inspectorate and issues with the compliance with biosecurity standards on pig farms. They also pointed out that the eradication of ASF among the wild boar population was not carried out in accordance with the procedures in force. Therefore, the Supreme Audit Office requested that the Minister of Agriculture and the Chief Veterinary Officer refine the regulations and modify ASF eradication plans [47].

## 5. The Role of Wild Boars and Humans in the Transmission of ASFV

As previously mentioned, wild boars are the primary vector of the virus in the natural environment, so migrations of these animals can have a significant role in spreading the virus distantly [19]. Recent studies confirm that the movement of wild boars contributes to the spread of the ASFV, but it is negligible. The rate of the spread of the disease among wild boars is slow due to the low mobility of these animals, their social and societal structure and the high virulence of the ASFV. The virus among feral pigs spreads gradually due to the continuity of the population of these animals rather than as a result of their long-distance migrations. Therefore, it is only local [48,49].

According to the European Food Safety Authority (EFSA), the average speed of ASFV transmission associated with the natural circulation of the virus in wild boar populations in Poland and the Baltic States ranges from 8 to 17 km per year [19]. Virus infection in these animals is influenced by population density and the size of forest complexes. ASF occurs mainly in wild boars in areas where the animal population exceeds 1 animal/km^2^. Therefore, the conclusion has been made that reducing the density of feral pigs below the above-mentioned values can limit the spread of the disease [50,51,52]. However, according to some other experts, even at low animal densities, i.e., below 0.1 boar/km^2^, the ASF virus can still spread in the environment through dead feral pigs killed by the disease [13].

## 6. Opole Voivodeship

A good example illustrating the way the ASF virus spreads in the environment is the region of Opole Voivodeship, where the populations of wild boar and domestic pigs had been disease-free until 2022. 

On 21 January 2022, positive cases were reported in dead boars near the village of Piaski, Brzeg County, the municipality of Lewin Brzeski. This led to the implementation of Zone II in the area as laid down by the European Commission. The monitoring activities conducted by hunters and the District Veterinary Inspectorate in Brzeg did not achieve the desired outcome—the disease was still spreading, as indicated by further positive cases in feral pigs. One of the contributing factors was the presence of many forest complexes in the region, which are inhabited by a large wild boar population.

In June 2022, another positive case was detected in the neighbouring Namysłów County. The height of the crops and the ubiquitous maize and rapeseed fields, which frequently covered 100 ha and were directly adjacent to forest complexes, made it impossible to efficiently reduce the number of wild boars. Moreover, it meant the risk of the disease entering local pig farms became very high. Further positive cases of shot wild boars in Namysłów County led to the decision to carry out dog searches in the infected area.

From September 2022, regular monitoring activities took place in the areas where positive cases were reported and the neighbouring forest complexes. The search areas were selected based on the analysis of the forest complexes themselves as well as the type of cultivation with the cooperation of the Chief District Hunter, whose expertise was invaluable. The two dogs used in the search activities were not hunting dogs. It was a deliberate decision not to use hunting breeds to eliminate the strong hunting instinct that they possess, which could have compromised the search operation in the event of an encounter with a wild animal. As a result, the search was more efficient. Each dog can cover 120–180 ha of forest per day; this allows a swift search of the infected area (Figure 5).

The number of positive cases in the shot wild boars did not reflect the grim situation that was witnessed by the search teams in the forests. During every search, the carcasses of feral pigs at various levels of decay were found.

When the first ASF cases were reported in Namysłów County and Opole County (both bordering Brzeg County), the Chief Veterinary Officer in Opole made the decision to introduce an odour-repellent barrier in the three counties. The perimeter was 100 km long and ran along forest and field tracks as well as highways and railway tracks. The A4 motorway was selected as a natural barrier in the south, where all the existing animal passes were closed off. A coordinated spraying operation was launched, covering the designated tracks, highways and railways tracks and repeated every 4 weeks for 6 months. The outcome was very optimistic—the spread of the disease in wild boars was, to a large extent, halted. Up until November 2022, there was not a single ASF case detected on the other side of the barrier. This applies to both shot and fallen (mainly in car accidents) wild boars. It is worth noting that driven hunts in Poland, by law, take place between November and March. Maize cultivation, which attracts wild boars, is over by then. Consequently, singulars move back into forests. This, combined with an exceptionally prolific wild mushroom season, which brought crowds into the forests, was the likely cause of the breach of the odour-repellent barrier in place.

Moreover, most probably, the growing population of wolves in Brzeg County and Namysłów County also contributed to this. The area provides fantastic hunting and breeding conditions that appeal to wolves. We think that when disturbed by wolves, wild boar sounders, especially when they have their young, tend to actively search for new feeding grounds. All this led to the first positive case being detected in November 2022 on the other side of the barrier in Namysłów County. Despite an intensive sanitary and scheduled shooting, the disease began to spread into new areas (Figure 6). 

An example of good practice and liaison between veterinary authorities, farmers and hunting associations is the efficient work of the District Veterinary Inspectorate in Namysłów in hunting zone no 27. The maize fields adjacent to the forests covered 230 ha in total. The hunters reported a problem with conducting sanitary shootings, arguing that wild boars seemed to constantly reside in the maize. The Veterinary Inspectorate deployed an uncrewed aerial vehicle (UAV) to carry out a search, which confirmed traces of wild boar activity on the maize fields. All the blood samples taken from the boars shot in the area tested positive for ASF (PCR test). To avoid the risk to humans and animals that a driven hunt in a maize field would pose, an urgent meeting was scheduled with the landowner and the manager of the field. The hazards connected with such a high number of potentially infected boars among the maize cultivated for fodder were discussed. As a result, the owner made the decision to cut the maize in the form of 6 m wide perpendicular lines. On those clear tracks, hunters placed traps, which efficiently reduced the population of wild boar on the field. The cut plants were used as a base for silage, whereas the rest of the harvest underwent a drying process and was subsequently quarantined in a separate silo to reduce the risk of contaminating farms.

By the end of November 2022, the aforementioned dogs found 40 boars in Namysłów County, 25 of which were diagnosed with ASF after a PCR test. Out of over 1000 shot boars in that year, only 8 were infected (Figure 7).

The mild winter and lack of significant snowfall facilitated monitoring and search activities. Wild boars were found in the eastern parts of Namysłów County and Brzeg County. Also, in Opole County, new outbreaks were reported. All the Veterinary Inspectorate workforce focused on the education of hunters, inspection and education of livestock farm owners and, first and foremost, on the search for and disposal of fallen boars. The efficient monitoring activities as well as prompt collection and destruction of carcasses ensured that the disease did not enter domestic pig farms. No ASF outbreak has been detected in domestic pigs in Opole Voivodeship. The aim of the intensification of search activities in the eastern region of Namysłów County is as swift and efficient as possible reduction in the disease in the environment, which is crucial in terms of minimising the spread. The objective is to prevent the disease from spreading to the eastern parts of Opole Voivodeship, especially to Kluczbork County and Olesno County, where the concentration of domestic pig farms is the highest in the region.

Another way of containing the disease, which is to be implemented in Opole Voivodeship, is the combination of a thorough inspection of the biosecurity of pig premises and risk assessment with regard to existing ASF outbreaks in wild boars. This combined with the analysis of planned cultivation and the knowledge of wild boar aetiology should help to identify the most ASF-prone fields, which will be treated with an odour repellent to prevent the contamination of crops and farming equipment and as a result, to protect local pig farms against the disease.

The proposed repellent works in two ways: based on wolf pheromones, it repels wild boars, at the same time discouraging them from consuming the treated plants, as it contains silicon, which causes strong salivation and an unpleasant taste in the oral cavity.

In March 2023, the first positive case was detected on the other side of the A4 motorway, which up until then had been considered an impenetrable barrier to the pathogen. An inspection concluded that the barricades used to block the animal passes had been damaged and animals were now able to cross from one side to the other. So far, this seems to be a single case. Monitoring activities in the area have not confirmed any fallen boars and none of the shot animals have been infected. 

As of the end of July 2023, in the whole of Opole Voivodeship, 328 positive cases of wild boars and 184 outbreaks have been reported. In total, 256 of those cases are fallen boars, and only 62 are the shot ones. This highlights the importance of efficient monitoring and search activities and prompt elimination of pathogens from the environment.

The fundamental indirect mechanism of ASFV transmission is contact between healthy feral pigs and infected carcasses of dead siblings [18,53,54,55,56]. The direct route of ASFV infection, unlike the indirect route, requires a high population density of wild boars for the disease to persist in the population. The contact of healthy animals with carrion, the specific carrier of the virus, can condition the spread of the disease at low wild boar numbers. In areas with high ASF mortality, the intensive search and disposal of fallen wild boars is recommended. Disease eradication programs recommend increasing the intensity of the search for dead wild boars or sanitary shooting of animals for rapid disposal of carcasses [57,58].

The issue facilitating the spread of ASF is not only the carcasses of dead boars but also contaminated soil or plants, so the ground on which the fallen boar lay should be disinfected to reduce the possibility of virus transmission [59].

Studies conducted in the Netherlands [60] have shown that the animals carrying the pathogen have a significant role in the spread of the virus. Scientists have proven that clinically healthy pigs that have survived an ASF infection (carriers) can transmit the virus to other pigs via direct contact. Based on this premise, it has been inferred by analogy that feral carriers can also infect other individuals with whom they come into contact [41].

According to EFSA experts, humans play the biggest part in the transmission of ASFV. Their activity may be the reason for the introduction of the virus into other areas and the spread of the disease over long distances [61,62]. The people using forest areas, which are the habitats of ASF-infected wild boars, can cause the spread of the virus since their clothing or footwear may be contaminated by it. The feeding of wild animals should be restricted in affected areas. In addition, hunting wild boars as a tool to combat ASF disease also carries the potential risk of contaminating the environment with this virus.

The research provided by this study confirms the previously presented thesis that humans play the biggest part in the long-distance transmission of ASFV. In 10 counties in Poland (Lubaczów, Parczew, Łosice, Bielsk podlaski, Lubartów, Mońki, Radzyń, Siemiatycze, Siedlce, Chełm), ASF outbreaks in domestic pigs occurred earlier than in the wild boar population. Moreover, in Zambrów and Wysokie Mazowieckie counties, ASF outbreaks were recorded only in pig farms [39].

## 7. Conclusions

The 9-year ASF epidemic in Poland shows that it is an interdisciplinary issue linked to the legal sector and the economic sector. The need to slaughter and utilise pigs, which if conducted on a large scale, can seriously affect smaller backyard farms. It is anticipated that ASFV will likely enter new areas. Therefore, it is imperative to follow the biosecurity measures and the government recommendations to combat ASF in Poland. The previous data shows that the number of wild boars infected with the virus is decreasing with a continuous increase in the number of animals with specific ASFV antibodies, which may indicate the beginning of ASF endemicity in Poland.

## Figures and Tables

**Figure 1 viruses-15-02325-f001:**
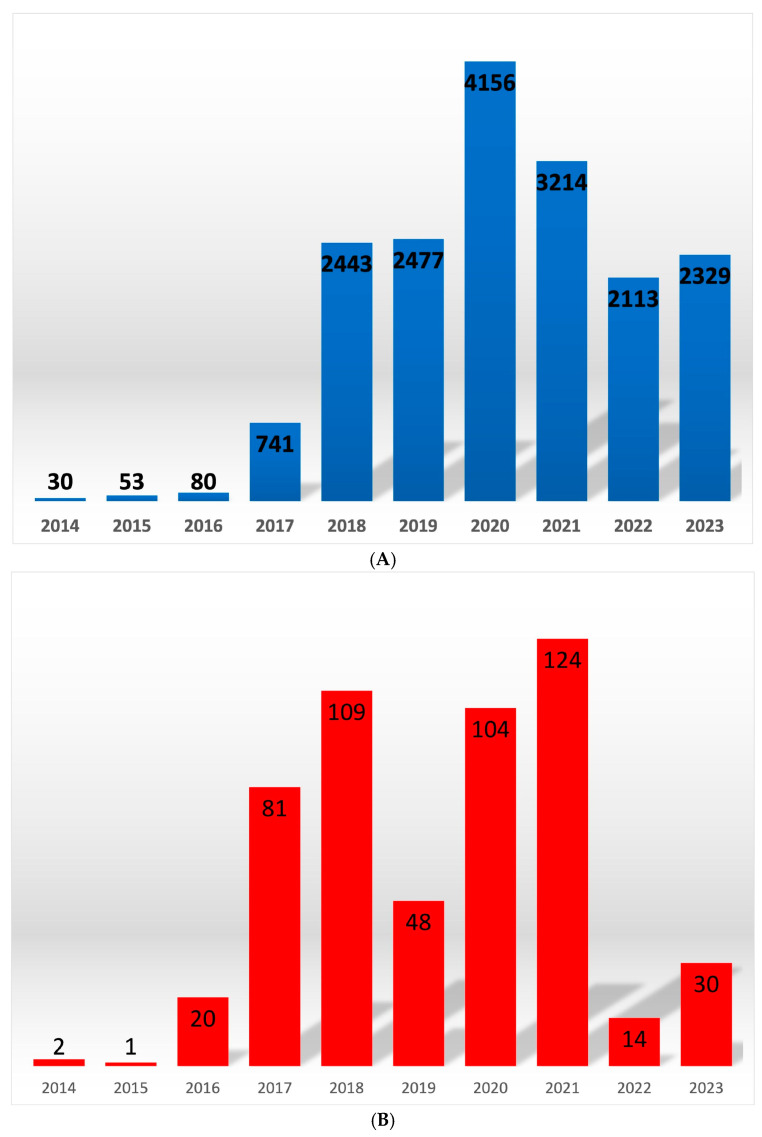
Number of ASF outbreaks in wild boar (**A**) and domestic pigs (**B**) during nine years of the epizootic.

**Figure 2 viruses-15-02325-f002:**
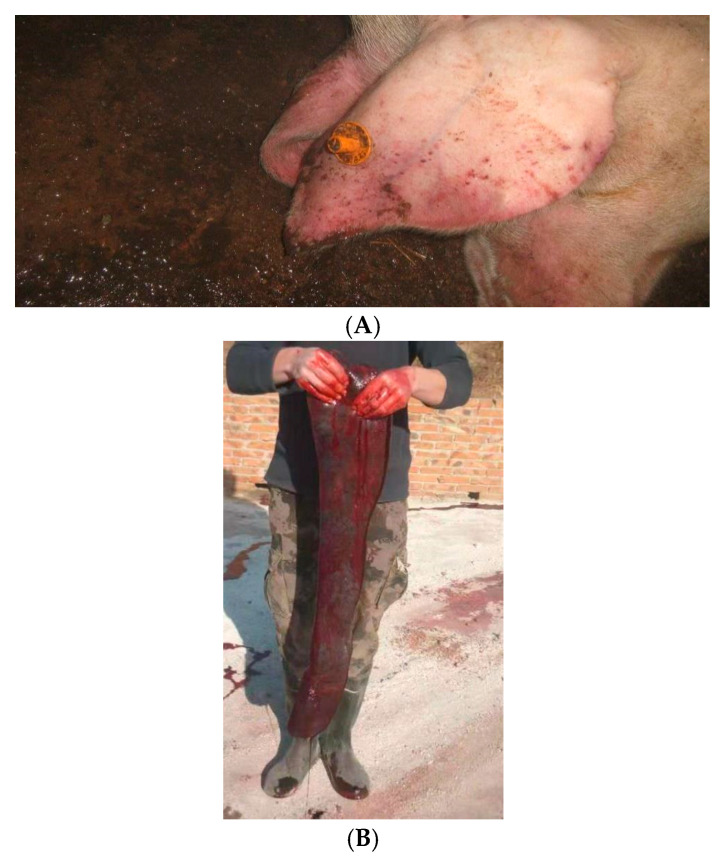
Clinical signs and most common lesions of ASF in domestic pigs. Cyanosis of the ears (**A**). Post-mortem splenomegaly (**B**).

**Figure 3 viruses-15-02325-f003:**
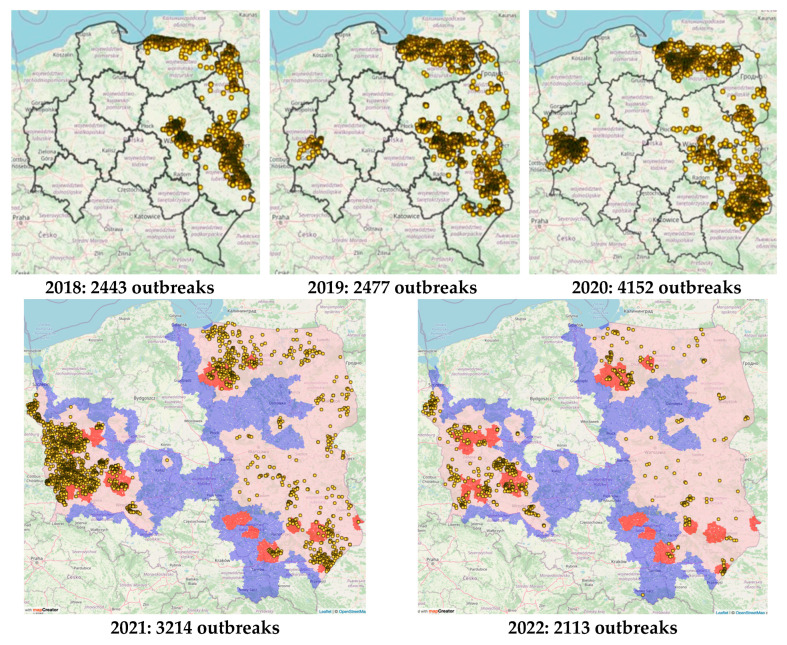
The peak of ASF expansion to distant locations in Poland between 2018 and 2020. The particular names of counties and voivodships in Polish are presented. The yellow dots presents outbreaks of ASF in wild boar. The zoning regulations are in place in accordance with the EU Decree 2023/2469 of 31 October 2023.

**Figure 4 viruses-15-02325-f004:**
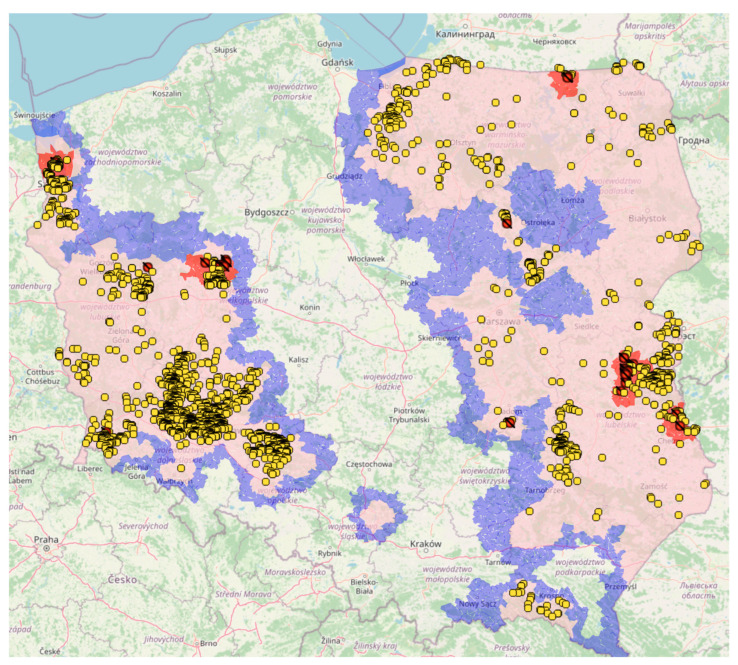
ASF outbreaks in wild boar (yellow points) and in domestic pigs (red dots) in 2023 in Poland. The zoning regulations are in place in accordance with the EU Decree 2023/2469 of 31 October 2023.

**Figure 5 viruses-15-02325-f005:**
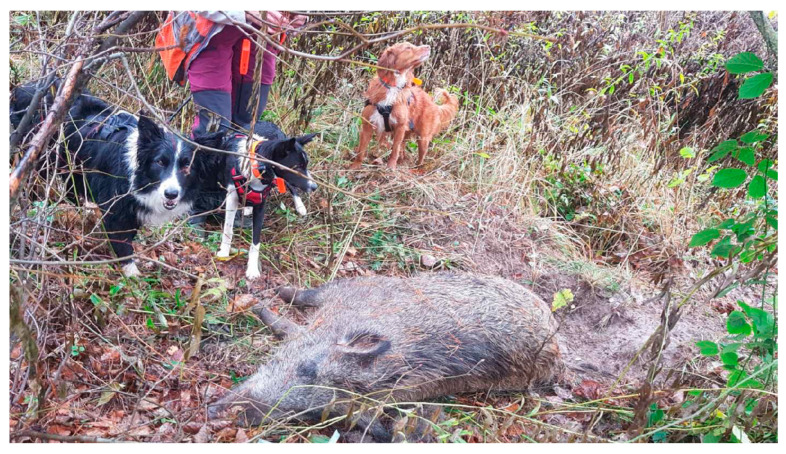
Search activities aimed at finding wild boar carcasses with the assistance of trained dogs are the most successful measure against the high contamination of the environment via ASFV. The best breeds of dogs to perform the searches are Border Collie and Toller.

**Figure 6 viruses-15-02325-f006:**
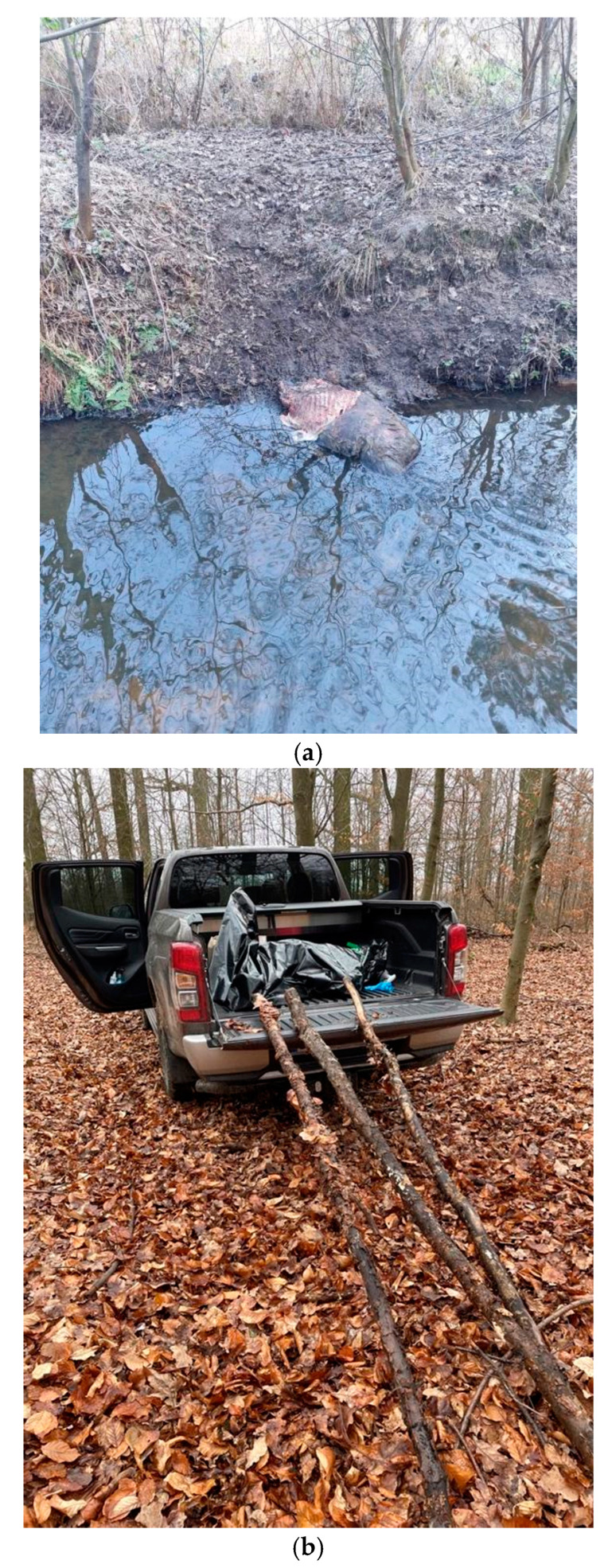
The location of a wild boar carcass may complicate effective disposal. (**a**) A common location of dead wild boar collection is related to water bodies (**b**) The disposal of a wild boar carcass is not possible without 4 × 4 vehicles, (**c**) The removal of a wild boar carcass from marshy areas could be performed using a quad bike.

**Figure 7 viruses-15-02325-f007:**
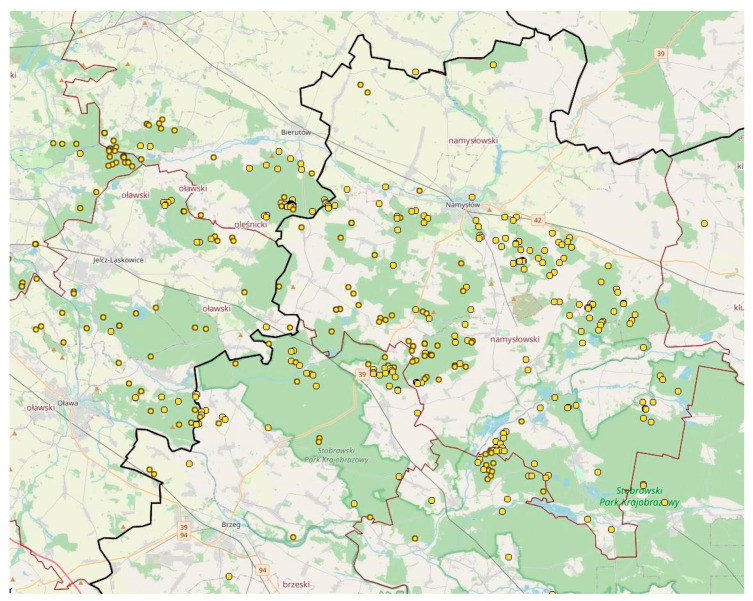
A map showing a location and scattering of new ASF outbreaks in wild boar in Opole voivodeship.The yellow points indicate recent outbreaks in wild boar.

## Data Availability

Not applicable.

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
