# Peer review of "Nine Years of African Swine Fever in Poland"

_viruses, 2023, doi:10.3390/v15122325_

Round 1
Reviewer 1 Report
Comments and Suggestions for Authors
African swine fever (ASF) is a highly contagious disease of pigs, and outbreaks cause tremendous impact on global swine production. The presented review summarizes the history and current situation of ASF in wild boars and domestic pigs in Poland. In general, this manuscript is well written and the experience in ASF controlling is might be useful for increase the knowledge and improvement of the control strategies against ASF.
Minor recommendations:
1. How is the population and density of domestic pigs and wild boars in Poland?
2. The authors are recommended to visualize the ASF outbreak events in Poland using a map.
3. The authors are suggested to discuss the global situation of ASF.
4. Lines 52-55: Maintaining high standards of biosecurity measures including disinfection may reduce many potential sources of infection Ticks of the genus Ornithodoros in endemic areas have been shown to be a frequent vector of the virus, participating in its transmission.
Before the word “Ticks”, ending punctuation was missing.
5. Please confirm the description is correct or not in Lines 60-61: ASFV is rapidly spreading across different parts of Europe and Asia as it has been confirmed in…
Author Response
Reviewer #1
Reviewer #1 African swine fever (ASF) is a highly contagious disease of pigs, and outbreaks
cause tremendous impact on global swine production. The presented review summarizes the
history and current situation of ASF in wild boars and domestic pigs in Poland. In general,
this manuscript is well written and the experience in ASF controlling is might be useful for
increase the knowledge and improvement of the control strategies against ASF.
Author GW: Many thanks for your kind and valuable support to improve our manuscript I
believe the changes introduced into the manuscript will meet your expectations. The changes
are marked in yellow.
Minor recommendations:
Reviewer #1: How is the population and density of domestic pigs and wild boars in Poland?
Author GW: The population of domestic pigs in Poland in 2023 was 8,3 mln of pigs
distributed in 50 thousand of pig farms. The average size of herd is 165 pigs. In case of wild
boar, the highest density over 2-4 wild boar per square km is in western Poland. The overall
estimated wild boar number is at the level of 67 thousand of wild boar. Additional comment
has been added in Introduction section line #83-86.
Reviewer #1: The authors are recommended to visualize the ASF outbreak events in Poland
using a map.
Author GW: Additional graphs showing number of ASF outbreaks in wild boar and domestic
pigs have been added. Additional map showing current epizootic situation in Poland during
2023 has been added.
Reviewer #1: The authors are suggested to discuss the global situation of ASF.
Author GW: The global situation has been discussed in introduction section between #67 and
#81 lines. Additional comments have been added also in lines#81-82.
Reviewer #1: Lines 52-55: Maintaining high standards of biosecurity measures including
disinfection may reduce many potential sources of infection Ticks of the genus Ornithodoros
in endemic areas have been shown to be a frequent vector of the virus, participating in its
transmission. Before the word “Ticks”, ending punctuation was missing.
Author GW: thank you for this remark an additional correction has been added.
Reviewer #1: Please confirm the description is correct or not in Lines 60-61: ASFV is rapidly
spreading across different parts of Europe and Asia as it has been confirmed in…
Author GW: This has been a misunderstanding. Additional corrections have been added in
lines #60-63.
Thank you for your valuable comments.
Reviewer 2 Report
Comments and Suggestions for Authors
A manuscript describing the course of events as regards ASF transmission in the last nine years in Poland. Parts of the manuscript are presented as a chronology and especially for readers who are not aware of the countries districts, it is quite difficult to follow the route of transmission among prefectures, as well as the zones created. Comments and corrections can be found in the attached pdf file. Moreover particular information that should be inserted includes:
· Maps of the areas affected presenting the gradual outbreaks expansion in wild and domestic population should be added.
· Selected tissues sampled and results would be also informative.
· Clinical signs and necropsy findings which occurred predominantly (eg as a table)
· Number of culled animals per year/region
· Estimation of financial losses/ per year/region
· Which counteracting measures taken in the past would be considered as partially efficacious and which other would be suggested as of minimal or no efficacy.
· The failure of biosecurity implementation in intensive farms (seen also elsewhere) and backyard farms, would be helpful to be further analyzed/discussed.

Particular typos have been highlighted as comments in the above-mentioned pdf file. Further check of English language is encouraged.
Author Response
Reviewer #1
Reviewer #1 African swine fever (ASF) is a highly contagious disease of pigs, and outbreaks
cause tremendous impact on global swine production. The presented review summarizes the
history and current situation of ASF in wild boars and domestic pigs in Poland. In general,
this manuscript is well written and the experience in ASF controlling is might be useful for
increase the knowledge and improvement of the control strategies against ASF.
Author GW: Many thanks for your kind and valuable support to improve our manuscript I
believe the changes introduced into the manuscript will meet your expectations. The changes
are marked in yellow.
Minor recommendations:
Reviewer #1: How is the population and density of domestic pigs and wild boars in Poland?
Author GW: The population of domestic pigs in Poland in 2023 was 8,3 mln of pigs
distributed in 50 thousand of pig farms. The average size of herd is 165 pigs. In case of wild
boar, the highest density over 2-4 wild boar per square km is in western Poland. The overall
estimated wild boar number is at the level of 67 thousand of wild boar. Additional comment
has been added in Introduction section line #83-86.
Reviewer #1: The authors are recommended to visualize the ASF outbreak events in Poland
using a map.
Author GW: Additional graphs showing number of ASF outbreaks in wild boar and domestic
pigs have been added. Additional map showing current epizootic situation in Poland during
2023 has been added.
Reviewer #1: The authors are suggested to discuss the global situation of ASF.
Author GW: The global situation has been discussed in introduction section between #67 and
#81 lines. Additional comments have been added also in lines#81-82.
Reviewer #1: Lines 52-55: Maintaining high standards of biosecurity measures including
disinfection may reduce many potential sources of infection Ticks of the genus Ornithodoros
in endemic areas have been shown to be a frequent vector of the virus, participating in its
transmission. Before the word “Ticks”, ending punctuation was missing.
Author GW: thank you for this remark an additional correction has been added.
Reviewer #1: Please confirm the description is correct or not in Lines 60-61: ASFV is rapidly
spreading across different parts of Europe and Asia as it has been confirmed in…
Author GW: This has been a misunderstanding. Additional corrections have been added in
lines #60-63.
Thank you for your valuable comments.
Reviewer #2
Reviewer #2: A manuscript describing the course of events as regards ASF transmission in
the last nine years in Poland. Parts of the manuscript are presented as a chronology and
especially for readers who are not aware of the country’s districts, it is quite difficult to follow
the route of transmission among prefectures, as well as the zones created. Comments and
corrections can be found in the attached pdf file. Moreover, particular information that should
be inserted includes:
Reviewer #2: Maps of the areas affected presenting the gradual outbreaks expansion in wild
and domestic population should be added.
Author GW: Additional maps showing evolution of ASFV spread in Poland have been added.
Additional graphs have been also presented. (Fig 1 and 2).
Reviewer #2: Selected tissues sampled and results would be also informative. Clinical signs
and necropsy findings which occurred predominantly (eg as a table)
Author GW: Additional figure showing the most important lesions in pigs have been added .
Due to the numerous outbreaks in pigs the exact presentation of clinical picture in all herds is
not possible.
Reviewer #2: Number of culled animals per year/region. Estimation of financial losses/ per
year/region
Author GW: These aspects are also difficult to estimate. Because of lack of direct database
from the General Veterinary Inspectorate.
Reviewer #2: Which counteracting measures taken in the past would be considered as
partially efficacious and which other would be suggested as of minimal or no efficacy.
Author GW: Additional measures have been discussed within the corrected paper.
Reviewer #2: Particular typos have been highlighted as comments in the above-mentioned pdf
file. Further check of English language is encouraged.\
Author GW: The manuscript has been proof-read by native speaking person. Thank you.
Reviewer #2: The failure of biosecurity implementation in intensive farms (seen also
elsewhere) and backyard farms, would be helpful to be further analyzed/discussed.
Author GW: The suggested issues have been also discussed within the corrected version.
Reviewer 3 Report
Comments and Suggestions for Authors
I reviewed the manuscript. However, the problems with word choices, sentence structure, paragraph structure, grammar, and punctuation make the manuscript impossible to review.
There is good information in this paper, and it will be interesting to readers. But the manuscript needs a thorough editing and rewrite of many sections, particularly the introduction and discussion sections. These changes are necessary to make this a useful article for an international audience.
Comments on the Quality of English Language
I reviewed the manuscript. However, the problems with word choices, sentence structure, paragraph structure, grammar, and punctuation make the manuscript impossible to review.
There is good information in this paper, and it will be interesting to readers. But the manuscript needs a thorough editing and rewrite of many sections, particularly the introduction and discussion sections. These changes are necessary to make this a useful article for an international audience.
Author Response
Reviewer #3
I reviewed the manuscript. However, the problems with word choices, sentence structure,
paragraph structure, grammar, and punctuation make the manuscript impossible to review.
There is good information in this paper, and it will be interesting to readers. But the
manuscript needs a thorough editing and rewrite of many sections, particularly the
introduction and discussion sections. These changes are necessary to make this a useful article
for an international audience.
Author GW: Dear Reviewer many thanks for your valuable remarks. The language issues
have been corrected in the current version of the manuscript.
Round 2
Reviewer 3 Report
Comments and Suggestions for Authors
This version of the manuscript is much improved and is ready for the final review by the journal's editorial staff. The content is descriptive and very interesting.
I would advise checking the use of the term "outbreak" - do the authors mean "case".
Comments on the Quality of English LanguageGood improvement